# Revisiting Overestimation Bias of Q-learning: Breaking Bias Propagation Chains Does Well

## Abstract

This paper revisits the overestimation bias of Q-learning from a new perspective, i.e., the breaking bias propagation chains. We make five-fold contributions. First, we analyze the estimation bias propagation chains of Q-learning, and find that the bias propagated from previous steps dominates the maximum Q-value estimation bias and slows the convergence speed, instead of the current bias. Second, we propose a novel positive-negative bias alternating algorithm called Alternating Q-learning (AQ). It breaks the unidirectional estimation bias propagation chains via alternately executing Q-learning and Double Q-learning. We show theoretically that there exist two suitable alternating parameters to eliminate the propagation bias. Third, we design an adaptive alternating strategy for AQ, obtaining Adaptive Alternating Q-learning (AdaAQ). It applies a softmax strategy with the absolute value of TD error to choose Q-learning or Double Q-learning for each state-action pair. Fourth, we extend AQ and AdaAQ to the large-scale settings with function approximation, i.e., including both discrete- and continuous-action Deep Reinforcement Learning (DRL). Fifth, both discrete- and continuous-action DRL experiments show that our method outperforms several baselines drastically; tabular MDP experiments reveal fundamental insights into why our method can achieve superior performance.

## 1 Introduction

As one of the most fundamental reinforcement learning algorithms, Q-learning Watkins et al. (1989) has been successfully applied to many real-world applications Zong et al. (2025); Arvanitidis & Alamaniotis (2024); Gao (2024) due to its simplicity and convergence guarantees under some mild assumptions Kearns & Singh (1998). However, Q-learning suffers from overestimation bias Thrun & Schwartz (1993), which can be exacerbated in DRL with nonlinear function approximation Mnih et al. (2015). This issue originates from the fact that the maximum Q-value is obtained by maximizing the stochastic estimations of Q-value. These stochastic estimations are caused by stochastic and unknown reward and state transition functions. Notably, the deadly triad Van Hasselt et al. (2018); Sutton et al. (2018) illustrates that the overestimation bias of Q-learning can be iteratively propagated via bootstrapping. Although this bias propagation phenomenon is well-known, most works Van Hasselt (2013); Peer et al. (2021); Schmitt-Förster & Sutter (2024); Tan et al. (2024c) focus on mitigating the overestimation bias instead of eliminating the bias propagation chains. For example, Double Q-learning Hasselt (2010) removes the overestimation bias via cross-validation, but may lead to underestimation bias. However, this underestimation bias can cause a slower learning speed and a larger performance penalty than the overestimation bias Ren et al. (2021); Li et al. (2023).

This paper provides a new perspective, i.e., the breaking bias propagation chains, to study the overestimation bias of Q-learning. In Section 3.2, we first set the asynchronous Q-value of Bellman optimality equation Silver (2015) as the ground truth. Then, we analyze the maximum Q-value estimation bias of Q-learning as Theorem 1, which consists of the current bias and the propagation bias. Notably, Corollary 1 demonstrates that the propagation bias rather than the current bias dominates the maximum Q-value estimation bias. That is to say, the propagation bias is the primary cause for the slow convergence speed. Example in Section 3.3 further illustrates that the ratio of propagation bias to current bias can be as high as 5 just after running Q-learning for four steps.

Based on the above analysis, this paper proposes a novel algorithm to break the unidirectional estimation bias propagation chains, called Alternating Q-learning. The main idea is to alternate between positive bias algorithms, such as Q-learning, and negative bias algorithms, such as Double Q-learning. Theorem 2 in Section 4.1 demonstrates that there exist two suitable alternating parameters to eliminate the propagation bias. To address the challenge in Alternating Q-learning, i.e., the optimal alternating parameters are unknown in advance and dynamically for different state-action pairs, we propose an adaptive alternating strategy, resulting in Adaptive Alternating Q-learning. It applies a softmax strategy with the absolute value of TD error to determine whether Alternating Q-learning should execute Q-learning or Double Q-learning for each state-action pair. Extensive experiment results show that our method outperforms several baselines drastically in tabular MDP, discrete-action DRL, and continuous-action DRL settings. In summary, this paper studies the overestimation bias of Q-learning from the breaking bias propagation chains perspective, and makes five key contributions as outlined in the abstract.

## 2 Related work

### 2.1 Underestimation bias methods

Double Q-learning Hasselt (2010); Van Hasselt (2013) is one notable algorithm, which uses cross-validation to decouple the maximum Q-value estimation. This decoupling process is achieved by maintaining two independent Q-tables: one Q-table is used to select the optimal action that attains the maximum Q-value; the other Q-table is used to estimate the Q-value associated with the previously selected optimal action. As a result, Double Q-learning removes the overestimation bias of Q-learning, but may lead to underestimation bias. EBQL Peer et al. (2021) is a natural extension of Double Q-learning to ensembles, and the estimation bias of maximum Q-value is always negative. REDQ Chen et al. (2021) reduces the estimation bias via a minimum operation over multiple random Q-tables, and the default size of random subset is two. But it still maintains a negative bias throughout most rounds of learning.

### 2.2 Control estimation bias methods

Weighted Double Q-learning Zhang et al. (2017) is a weighted combination of Q-learning and Double Q-learning, and controls the estimation bias through the weight parameter. Averaged Q-learning Anschel et al. (2017) averages multiple independent Q-tables to reduce the variance of Q-values, and finds that the estimation bias is inversely proportional to the number of Q-tables. With a finite number of Q-tables, the estimation bias of Averaged Q-learning is always positive. Softmax Q-learning Song et al. (2019) demonstrates that the estimation bias is proportional to the hyperparameter of softmax operation. Maxmin Q-learning Lan et al. (2020) uses a minimum operation over multiple independent Q-tables, and finds that the estimation bias is inversely proportional to the number of Q-tables. AdaEQ Wang et al. (2021) adjusts the ensemble size of Maxmin Q-learning with the approximation Q-value error to control the estimation bias, note that this adjustment method relies on the discounted MC return Li (2023). Self-Correcting Q-learning Zhu & Rigotti (2021) builds a self-correcting estimator with the current and last Q-values, and controls the estimation bias via dynamically adjusting the Pearson correlation coefficient between successive iterations. Balanced Q-learning Karimpanal et al. (2023) computes the optimistic and pessimistic biases with the maximum and minimum operations, respectively, and balances them to control the estimation bias via the balancing factor. AEQ Gong et al. (2023) uses the uncertainty of Q-values and the familiarity of sampling trajectories to control the estimation bias. AdaOrder Q-learning Tan et al. (2024c) uses the order statistic of multiple independent Q-tables to control the estimation bias, which can satisfy the fine-grained bias needs for different environments.

The maximum expected Q-value is impossible to compute without the underlying state transition probabilities and reward distributions Ishwaei D et al. (1985). Thus, there exist estimation bias and its propagation via bootstrapping. However, existing methods focus on the current estimation bias, instead of the propagation bias. Different from previous methods, this paper revisits the propagation process of estimation bias in Section 3.2, and finds that even though previous estimation biases can be diluted during propagation, the propagation bias still dominates the maximum Q-value estimation bias and slows the convergence speed.

## 3 PROBLEM ANALYSIS

### 3.1 OVERESTIMATION BIAS OF Q-LEARNING

We consider an infinite-horizon MDP Chen et al. (2022), where each decision step is indexed by $t \in \mathbb{N}$. Let $\mathcal{S}$ and $\mathcal{A}$ denote the state space and the action space, respectively. Let $P\left(s'|s,a\right)$ denote the state transition probability under $(s,a) \in \mathcal{S} \times \mathcal{A}$, where $s' \in \mathcal{S}$ is the next state. Let $R\left(s,a\right)$ denote the reward associated with $(s,a)$. Let $s \mapsto \pi\left(s\right)$ denote the policy, where $s \in \mathcal{S}$ and $\pi\left(s\right) \in \mathcal{A}$. Let $Q_\pi\left(s,a\right)$ denote the expected cumulative discounted reward with discounting factor $\gamma \in (0,1)$ as:

$$Q_\pi\left(s,a\right) = \mathbb{E}\left[R\left(S_t,A_t\right) + \gamma R\left(S_{t+1},A_{t+1}\right) + \gamma^2 R\left(S_{t+2},A_{t+2}\right) + \cdots | S_t = s, A_t = a\right],$$

where $S_\kappa$ is generated from $P(S_\kappa|S_{\kappa-1},A_{\kappa-1})$ and $A_\kappa = \pi(S_\kappa)$, $\forall \kappa \geq t+1$. Note that $S_\kappa$ and $A_\kappa$ are the state random variable and the action random variable, respectively. The learning objective is to find the optimal Q-value and the optimal policy. More specifically, the optimal Q-value is $Q^*\left(s,a\right) = \max_{\pi \in \Pi} Q_\pi\left(s,a\right)$, where $\Pi$ denotes a set of all policy; the optimal policy is $\pi^*\left(s\right) = \arg\max_{a \in \mathcal{A}} Q^*\left(s,a\right), \forall\left(s,a\right) \in \mathcal{S} \times \mathcal{A}$.

The optimal Q-value is unknown in advance for model-free reinforcement learning. Q-learning maintains one Q-table $Q_t\left(s,a\right)$ in each time step $t$, and uses this Q-table to estimate the optimal Q-value. More specifically, at each time step $t$, Q-learning uses $\varepsilon$-greedy policy Rodrigues Gomes & Kowalczyk (2009) with $\arg\max_{a \in \mathcal{A}} Q_t\left(s,a\right)$, i.e., $\varepsilon \in (0,1)$, to interact with the environment, obtains the sample data $\{s,a,R\left(s,a\right),s'\}$ and updates the Q-table as:

$$Q_{t+1}\left(s,a\right) = Q_t\left(s,a\right) + \alpha\left[R\left(s,a\right) + \gamma \max_{a' \in \mathcal{A}} Q_t\left(s',a'\right) - Q_t\left(s,a\right)\right], \tag{1}$$

where $\alpha$ is the learning rate, $Y_t\left(s,a\right) = R\left(s,a\right) + \gamma\max_{a' \in \mathcal{A}} Q_t\left(s',a'\right)$ is the target Q-value. Under some mild conditions Kearns & Singh (1998), the estimated Q-value of Q-learning is guaranteed to converge to the optimal Q-value, i.e., $\lim_{t \to +\infty} Q_t\left(s,a\right) = Q^*\left(s,a\right), \forall\left(s,a\right) \in \mathcal{S} \times \mathcal{A}$.

We set the asynchronous Q-value, denoted by $\hat{Q}_t\left(s,a\right)$, of Bellman optimality equation as the ground truth Silver (2015), which updates as:

$$\hat{Q}_{t+1}\left(s,a\right) = \hat{Q}_t\left(s,a\right) + \alpha\left[\mathbb{E}\left[R\left(s,a\right) + \gamma\max_{a' \in \mathcal{A}}\hat{Q}_t\left(s',a'\right)\right] - \hat{Q}_t\left(s,a\right)\right], \tag{2}$$

where $\hat{Y}_t\left(s,a\right) = \mathbb{E}\left[R\left(s,a\right) + \gamma\max_{a' \in \mathcal{A}}\hat{Q}_t\left(s',a'\right)\right]$ is the unbiased target Q-value. We focus on the maximum operation of Q-learning, which is the root cause of overestimation bias. Following Thrun & Schwartz (1993), although we assume that $Q_t\left(s',a'\right)$ is an unbiased estimator for $\hat{Q}_t\left(s',a'\right), \forall a' \in \mathcal{A}$, according to Jensen's inequality Hansen & Pedersen (2003), we have: $\mathbb{E}\left[\max_{a' \in \mathcal{A}} Q_t\left(s',a'\right)\right] \geq \max_{a' \in \mathcal{A}}\mathbb{E}\left[Q_t\left(s',a'\right)\right] = \max_{a' \in \mathcal{A}}\hat{Q}_t\left(s',a'\right)$. Figure 1(a) also verifies that this overestimation bias can slow the convergence speed of Q-learning.

### 3.2 BIAS PROPAGATION CHAINS

For compactness, we write $Q_i, Y_i, \hat{Q}_i, \hat{Y}_i$ instead of $Q_i\left(s,a\right), Y_i\left(s,a\right), \hat{Q}_i\left(s,a\right), \hat{Y}_i\left(s,a\right), \forall\left(s,a\right) \in \mathcal{S} \times \mathcal{A}$. Then, we expand Equation (1) and Equation (2) as:

$$\begin{cases} Q_{t+1} = (1-\alpha)^{t+1} Q_0 + \sum_{i=0}^{t} \alpha(1-\alpha)^{t-i} Y_i, \\ \hat{Q}_{t+1} = (1-\alpha)^{t+1}\hat{Q}_0 + \sum_{i=0}^{t} \alpha(1-\alpha)^{t-i}\hat{Y}_i. \end{cases} \tag{3}$$

We set $Q_0 = \hat{Q}_0$ with the same initial Q-value; $e_i = Q_i - \hat{Q}_i$ as the Q-value estimation error; $Z_i = Y_i - \hat{Y}_i$ as the target Q-value estimation bias. Then, we have:

$$e_{t+1} = \sum_{i=0}^{t} \alpha(1-\alpha)^{t-i} Z_i. \tag{4}$$

To analyze the overestimation bias propagation chains of Q-learning, following Thrun & Schwartz (1993); Lan et al. (2020), we first make a common assumption as:

**Assumption 1** *The Q-value estimation error obeys a uniform distribution as:*

$$e_{t+1} \sim U\left(\mu_{t+1} - \xi_{t+1}, \mu_{t+1} + \xi_{t+1}\right),$$

*where* $\mathbb{E}\left[e_{t+1}\right] = \sum_{i=0}^{t} \alpha\left(1-\alpha\right)^{t-i} \mathbb{E}\left[Z_i\right] = \mu_{t+1} \geq 0$ *due to the positive bias of Q-learning;* $0 \leq \xi_{t+1} \leq \xi_t$ *due to the increasing number of samples and convergence guarantees. Note that* $\forall a \in \mathcal{A}$, *the Q-value estimation error* $e_{t+1}$ *are independent and identically distributed (i.i.d.).*

Following the target Q-value estimation bias, we have:

$$Z_{t+1} = R\left(s,a\right) - \mathbb{E}\left[R\left(s,a\right)\right] + \gamma\left(\max_{a'\in\mathcal{A}} Q_{t+1} - \mathbb{E}\left[\max_{a'\in\mathcal{A}} \hat{Q}_{t+1}\right]\right). \quad (5)$$

**Theorem 1** *Based on Assumption 1, the expected maximum Q-value estimation bias is as:*

$$\mathbb{E}\left[Z_{t+1}\right] = \underbrace{\gamma\xi_{t+1}\frac{|\mathcal{A}|-1}{|\mathcal{A}|+1}}_{cur\text{-}bias} + \underbrace{\gamma\sum_{i=0}^{t}\alpha\left(1-\alpha\right)^{t-i}\mathbb{E}\left[Z_i\right]}_{prop\text{-}bias}.$$

Theorem 1 is a generalization of the first Lemma in Thrun & Schwartz (1993); we provide the proof in Appendix A. Theorem 1 demonstrates that in Q-learning, the maximum Q-value estimation bias consists of two components: the current bias (cur-bias) and the propagation bias (prop-bias).

**Corollary 1** *When the discount factor* $\gamma \geq \frac{\xi_{t+1}}{\xi_t}$, *we have:*

$$\lim_{t\to+\infty} \frac{\gamma\sum_{i=0}^{t}\alpha\left(1-\alpha\right)^{t-i}\mathbb{E}\left[Z_i\right]}{\gamma\xi_{t+1}\frac{|\mathcal{A}|-1}{|\mathcal{A}|+1}} \geq 1.$$

We prove Corollary 1 in Appendix B. It illustrates that during Q-learning updates, the prop-bias progressively dominates the maximum Q-value estimation bias composition, instead of the cur-bias.

### 3.3 Example

Consider a simple multi-armed bandit setting Zhang et al. (2017), which includes one state $S$ with ten identical actions, and each action returns a reward following $\mathcal{N}(0,1)$. Following previous works Zhang et al. (2017); Tan et al. (2024c), we set $\gamma = 0.95$, $\alpha = 0.5$, $\varepsilon = \frac{1}{n(S)}$, $Q_0\left(S,a\right) \sim \mathcal{N}(0,1)$ for each action $a$, where $n\left(S\right)$ is the visited number of state $S$.

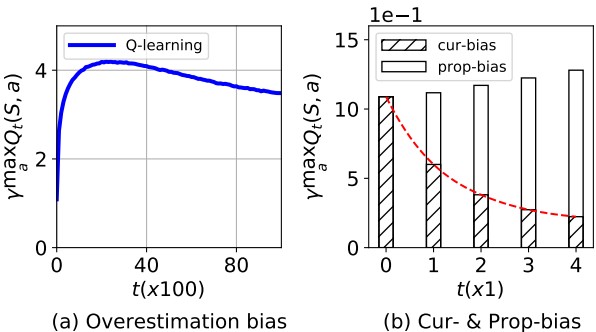

(a) Overestimation bias      (b) Cur- & Prop-bias

Figure 1: The discounted maximum Q-value of Q-learning.

Figure 1 shows the discounted maximum Q-value at state $S$, denoted by $\gamma\max_a Q_t\left(S,a\right)$, of Q-learning across steps $t$. All results are averaged over $10,000$ runs. Note that the maximum expected Q-value is zero. According to Theorem 1, we compute prop-bias as $\gamma\sum_{i=0}^{t-1}\alpha\left(1-\alpha\right)^{t-i-1}\max_a Q_i\left(S,a\right)$, and compute cur-bias as $\gamma\max_a Q_t\left(S,a\right) - \gamma\sum_{i=0}^{t-1}\alpha\left(1-\alpha\right)^{t-i-1}\max_a Q_i\left(S,a\right)$. Figure 1(a) shows that Q-learning has an overestimation bias, which slows convergence speed. Figure 1(b) shows that prop-bias accumulates the previous positive bias, and progressively dominates the maximum Q-value estimation bias composition. More specifically, the ratio of prop-bias to cur-bias is $\frac{1.06}{1.28-1.06} \approx 5$ at $t = 4$.

# 4 METHODS

## 4.1 ALTERNATING Q-LEARNING

Due to the overestimation bias of Q-learning, the prop-bias in Theorem 1 accumulates the unidirectional positive bias. Even though previous positive biases can be diluted during propagation, Corollary 1 shows that the prop-bias still dominates the maximum Q-value estimation bias. Example in Section 3.3 further illustrates that the prop-bias is the primary cause for the slow convergence speed in Q-learning. Similarly, Double Q-learning tends to accumulate the unidirectional negative bias, may lead to a slower convergence speed than Q-learning Ren et al. (2021); Li et al. (2023).

To break the above unidirectional estimation bias propagation chains and improve the convergence speed, we propose a novel positive-negative bias alternating execution framework. More specifically, this framework includes two hyperparameters $(M, N)$, where $M \in \mathbb{N}^+$ is the number of alternating execution steps for the positive bias algorithms, such as Q-learning Watkins et al. (1989) and Averaged Q-learning Anschel et al. (2017); $N \in \mathbb{N}^+$ is the corresponding steps for the negative bias algorithms, such as Double Q-learning Hasselt (2010) and EBQL Peer et al. (2021). Note that we set Q-learning and Double Q-learning as a pair of positive-negative bias algorithms, obtaining our Alternating Q-learning (AQ) as Algorithm 1.

---

**Algorithm 1 Alternating Q-learning**

1: **Parameter**: $M, N$
2: **Initialize**: $Q_0^1(s, a), Q_0^2(s, a), \forall (s, a) \in \mathcal{S} \times \mathcal{A}$
3: Get the starting state $s$
4: **for** $t = 0, 1, 2, \cdots$ **do**
5:     Choose action $a$ at state $s$ by $\varepsilon$-greedy policy with $\arg\max_{a \in \mathcal{A}} \frac{Q_t^1(s,a) + Q_t^2(s,a)}{2}$
6:     Take action $a$, get reward $R(s, a)$ and next state $s'$
7:     Randomly select one Q-table $k$ from $\{1, 2\}$ to update
8:     **if** $t \mod (M + N) < M$ **then**
9:         $Q_{t+1}^k(s, a) = Q_t^k(s, a) + \alpha \left[ R(s, a) + \gamma \max_{a' \in \mathcal{A}} Q_t^k(s', a') - Q_t^k(s, a) \right]$
10:     **else**
11:         $Q_{t+1}^k(s, a) = Q_t^k(s, a) + \alpha \left[ R(s, a) + \gamma Q_t^{3-k}\left(s', \arg\max_{a' \in \mathcal{A}} Q_t^k(s', a')\right) - Q_t^k(s, a) \right]$
12:     $s \longleftarrow s'$

---

AQ maintains two independent Q-tables $Q_t^1(s, a), Q_t^2(s, a)$ in each time step $t$. When $t \mod (M + N) < M$, AQ uses Q-learning to update, the target Q-value $Y_t^k = R(s, a) + \gamma \max_{a' \in \mathcal{A}} Q_t^k$; when $t \mod (M + N) \geq M$, AQ uses Double Q-learning to update, the target Q-value $Y_t^k = R(s, a) + \gamma Q_t^{3-k}\left(s', \arg\max_{a' \in \mathcal{A}} Q_t^k(s', a')\right)$. To ensure fair comparison, we also consider two Q-tables $\hat{Q}_t^1(s, a), \hat{Q}_t^2(s, a)$ for Bellman optimality equation as Equation (2), and the unbiased target Q-value $\hat{Y}_t^k = \mathbb{E}\left[R(s, a) + \gamma \max_{a' \in \mathcal{A}} \hat{Q}_t^k\right]$. Following Section 3.2, we set the Q-value estimation error as: $e_{t+1}^k = Q_{t+1}^k - \hat{Q}_{t+1}^k$; set the target Q-value estimation bias as: $Z_i^k = Y_i^k - \hat{Y}_i^k$; and have: $e_{t+1}^k = \sum_{i=0}^t \alpha(1 - \alpha)^{t-i} Z_i^k$, where $e_{t+1}^k \sim U\left(\mu_{t+1}^k - \xi_{t+1}^k, \mu_{t+1}^k + \xi_{t+1}^k\right)$. Note that the sign of $\mathbb{E}\left[e_{t+1}^k\right] = \mu_{t+1}^k$ is unknown in advance due to the alternating mechanism.

**Theorem 2** *Under the above statement, the expected estimation bias of AQ is as follows.*

- *When $t \mod (M + N) < M$, we have:*

$$\mathbb{E}\left[Z_{t+1}^k\right] = \underbrace{\gamma \xi_{t+1}^k \frac{|\mathcal{A}| - 1}{|\mathcal{A}| + 1}}_{\text{cur-bias}} + \underbrace{\gamma \sum_{i=0}^t \alpha(1 - \alpha)^{t-i} \mathbb{E}\left[Z_i^k\right]}_{\text{prop-bias}}.$$

- *When $t \mod (M + N) \geq M$, we have:*

$$\mathbb{E}\left[Z_{t+1}^k\right] = \underbrace{\gamma \sum_{j=1}^{|\mathcal{A}|} \mathbb{E}\left[\hat{Q}_{t+1}^k(s', a_j') - \hat{Q}_{t+1}^k(s', \hat{a}')\right] p\left(a_j' = \hat{a}'\right)}_{\text{cur-bias}} + \underbrace{\gamma \sum_{i=0}^t \alpha(1 - \alpha)^{t-i} \mathbb{E}\left[Z_i^k\right]}_{\text{prop-bias}}.$$

*Note that $\xi_{t+1}^k \geq 0$; $a_j' = \arg\max_{a' \in \mathcal{A}} Q_{t+1}^k(s', a')$; $\hat{a}' = \arg\max_{a' \in \mathcal{A}} \hat{Q}_{t+1}^k(s', a')$.*

We prove Theorem 2 in Appendix C. It shows that when $t \mod (M + N) < M$, AQ selects Q-learning to update, the cur-bias is positive, the prop-bias accumulates the positive bias; when $t \mod (M + N) \geq M$, AQ selects Double Q-learning to update, the cur-bias is negative, the prop-bias accumulates the negative bias. Therefore, AQ breaks the unidirectional estimation bias propagation chains via alternately executing Q-learning and Double Q-learning. More specifically, the estimation bias is proportional to $M$ and inversely proportional to $N$. When $M \to +\infty$ and $N = 1$, AQ gets the upper bound, but is always smaller than Q-learning; when $M = 1$ and $N \to +\infty$, AQ gets the lower bound, but is always larger than Double Q-learning. Due to that the cur-bias can vary between positive and negative, the prop-bias can theoretically be eliminated by two suitable parameters $(M, N)$.

## 4.2 ADAPTIVE ALTERNATING Q-LEARNING

AQ alternately executes Q-learning and Double Q-learning via $(M, N)$. However, the optimal $(M, N)$ are unknown in advance and dynamically for different state-action pairs. Therefore, we need to design an adaptive alternating strategy for AQ.

Following Algorithm 1, we first define the TD error Zhang et al. (2021) of Q-learning in each time step $t$ as: $td_t^Q(s, a) = R(s, a) + \gamma \max_{a' \in \mathcal{A}} Q_t^k(s', a') - Q_t^k(s, a)$; the corresponding TD error of Double Q-learning as: $td_t^{DQ}(s, a) = R(s, a) + \gamma Q_t^{3-k}(s', \arg\max_{a' \in \mathcal{A}} Q_t^k(s', a')) - Q_t^k(s, a)$. Although the TD error has inherent uncertainty during updates, the convergence guarantees of algorithms still support it as a reliable feedback signal for tracking the variation trend of maximum Q-value estimation bias. More specifically, a large $td_t^Q(s, a)$ reflects significant positive bias of Q-learning, and AQ should switch to Double Q-learning to suppress this bias and prevent its propagation from enlarging prop-bias. Similarly, when $td_t^{DQ}(s, a)$ is small, AQ should switch to Q-learning to counteract the significant negative bias of Double Q-learning. Thus, we apply a softmax strategy based on the absolute value of TD error to compute the alternating execution probabilities as:

$$\mathbb{P}_t^{(s,a)}[Q] = \frac{e^{\tau|td_t^{DQ}(s,a)|}}{e^{\tau|td_t^Q(s,a)|} + e^{\tau|td_t^{DQ}(s,a)|}}, \mathbb{P}_t^{(s,a)}[DQ] = \frac{e^{\tau|td_t^Q(s,a)|}}{e^{\tau|td_t^Q(s,a)|} + e^{\tau|td_t^{DQ}(s,a)|}}, \quad (6)$$

where $\tau \geq 0$ denotes the temperature parameter; $\mathbb{P}_t^{(s,a)}[Q]$ and $\mathbb{P}_t^{(s,a)}[DQ]$ represent the alternating probabilities to Q-learning and Double Q-learning, respectively.

Incorporating the above softmax strategy into AQ, we obtain Adaptive Alternating Q-learning (AdaAQ) as Algorithm 2. Note that at line 8, AQ selects Q-learning or Double Q-learning via categorical sampling.

---

**Algorithm 2 Adaptive Alternating Q-learning**

---

1: **Initialize**: $Q_0^1(s, a), Q_0^2(s, a), \forall (s, a) \in \mathcal{S} \times \mathcal{A}$
2: Get the starting state $s$
3: **for** $t = 0, 1, 2, \cdots$ **do**
4:      Choose action $a$ at state $s$ by $\varepsilon$-greedy policy with $\arg\max_{a \in \mathcal{A}} \frac{Q_t^1(s,a) + Q_t^2(s,a)}{2}$
5:      Take action $a$, get reward $R(s, a)$ and next state $s'$
6:      Randomly select one Q-table $k$ from $\{1, 2\}$ to update
7:      Compute $\mathbb{P}_t^{(s,a)}[Q], \mathbb{P}_t^{(s,a)}[DQ]$ as Equation (6)
8:      Select algorithm via categorical sampling: $AQ \sim \text{Cat}\left([Q, DQ], \left[\mathbb{P}_t^{(s,a)}[Q], \mathbb{P}_t^{(s,a)}[DQ]\right]\right)$
9:      **if** $AQ == Q$ **then**
10:          $Q_{t+1}^k(s, a) = Q_t^k(s, a) + \alpha\left[R(s, a) + \gamma \max_{a' \in \mathcal{A}} Q_t^k(s', a') - Q_t^k(s, a)\right]$
11:      **else**
12:          $Q_{t+1}^k(s, a) = Q_t^k(s, a) + \alpha\left[R(s, a) + \gamma Q_t^{3-k}(s', \arg\max_{a' \in \mathcal{A}} Q_t^k(s', a')) - Q_t^k(s, a)\right]$
13:      $s \longleftarrow s'$

---

### 4.3 EXTENSION TO DRL: DISCRETE- AND CONTINUOUS-ACTION SPACES

Following the previous DRL algorithms Silver et al. (2014); Mnih et al. (2015), we represent the Q-function by a neural network for high-dimensional environments, and try to extend our methods to DRL. More specifically, for the discrete-action DRL settings, such as Atari Bellemare et al. (2013), we set DQN Mnih et al. (2015) and DDQN Van Hasselt et al. (2016) as a pair of positive-negative bias algorithms, and extend AQ and AdaAQ to Alternating DQN (ADQN) as Appendix D.1 and Adaptive Alternating DQN (AdaADQN) as Appendix D.2, respectively; for the continuous-action DRL settings, such as Mujoco Brockman et al. (2016), we set DDPG Silver et al. (2014) and TD3 Fujimoto et al. (2018) as a pair of positive-negative bias algorithms, and extend AQ and AdaAQ to Alternating DDPG (ADDPG) as Appendix E.1 and Adaptive Alternating DDPG (AdaADDPG) as Appendix E.2, respectively.

## 5 EXPERIMENTS

### 5.1 TABULAR MDP EXPERIMENTS

**MDP environments:** (1) Multi-armed bandit is shown in Section 3.3. (2) Roulette is adapted from Lee & Powell (2019). Like Tan et al. (2024b), we simplify Roulette to 13 actions: six $2:1$ bets on 12 numbers (win 0.3158); six $1:1$ bets on 18 numbers (win 0.4737); one $1:1$ bet on nothing (win 0.5). (3) Gridworld $3 \times 3$ and $4 \times 4$ Zhu & Rigotti (2021) have four cardinal actions for each state, with start (southwest) and goal (northeast) positions. The agent resets to the start state upon reaching the goal, while boundary-violating actions maintain the current state. Non-goal states yield equiprobable stochastic rewards ($-12$ or $10$), while goal-state yields equiprobable stochastic rewards ($-30$ or $40$).

**Parameter settings:** Following Hasselt (2010); Pentaliotis & Wiering (2021); Tan et al. (2024a), we set $\gamma = 0.95$, $\varepsilon = \frac{1}{n(s)^{0.5}}$ by default; $\alpha = \frac{1}{n(s,a)^{0.8}}$ for Multi-armed bandit and Roulette; $\alpha = \frac{1}{n(s,a)^{1.0}}$ for Gridworld, where $n(s)$ and $n(s,a)$ are the visited number of state $s$ and state-action pair $(s,a)$, respectively. Note that all experiment results are averaged over $1,000$ runs.

**Comparison baselines:** We vary $M = 1, 2, 4, 8, 16$; $N = 1, 2, 4, 8, 16$ for our AQ; $\tau = 1, 2, 5, 10, 100$ for our AdaAQ, and set $\tau = 1$ by default. We consider eight comparison baselines as: Averaged Q-learning (AvgQ) Anschel et al. (2017), Maxmin Q-learning (MQ) Lan et al. (2020), Self-Correcting Q-learning (SCQ) Zhu & Rigotti (2021), Softmax Q-learning (SoftQ)Song et al. (2019), Weighted Double Q-learning (WDQ) Zhang et al. (2017), REDQ Chen et al. (2021), EBQL Peer et al. (2021), AdaEQ Wang et al. (2021). For a fair comparison, we set the number of Q-tables of AvgQ and MQ as 2; the temperature parameter of SoftQ as 1. For other baselines, we set the self-correcting parameter of SCQ as 2; the adaptive adjustment parameter of WDQ as 1; the number of Q-tables of REDQ, EBQL, AdaEQ as 10; which are recommended and fine-tuned.

Figure 2 shows the maximum Q-value, the probability of betting nothing denoted by $\Pr[leave]$, and the average reward per step of our methods in tabular MDP environments. **AQ:** Figure 2(a-b) show that the maximum Q-value of AQ is proportional to $M$ and inversely proportional to $N$, and it always lies between that of Q-learning and Double Q-learning. This implies that AQ can effectively break the unidirectional estimation bias propagation chains. **AdaAQ:** Figure 2(c-d) show that the maximum Q-value curves of AdaAQ with different $\tau$ overlap, and converge to zero faster than that of AQ. This implies that AdaAQ is not sensitive to $\tau$, and can provide an adaptive alternating strategy for AQ. **Comparison:** Figure 2(e-h) show that AdaAQ can estimate the maximum expected Q-value more accurately than the eight baselines, resulting in better policy and higher reward. Appendix F.1 provides the table of comparison results.

Table 1 shows the results of our AQ and AdaAQ with different parameters, where all values are evaluated in the final round. From the maximum Q-value perspective, AdaAQ(1), i.e., $4.14 \times 10^{-3}$, is one level improvement over AQ(4, 8), i.e., $-7.04 \times 10^{-2}$, in Multi-armed bandit. From the policy perspective, compared to AQ(4, 4), AdaAQ(100) improves by $93.59\% - 87.43\% = 6.16\%$ in Roulette. From the reward perspective, AdaAQ(10) outperforms AQ(4, 4) by $\frac{-0.12-(-0.492)}{0.492} = 75.60\%$ in Gridworld $3 \times 3$; AdaAQ(5) outperforms AQ(4, 4) by $\frac{-0.476-(-0.73)}{0.73} = 34.79\%$ in Gridworld $4 \times 4$. Appendix F.2 provides the figure of our AQ and AdaAQ with different parameters.

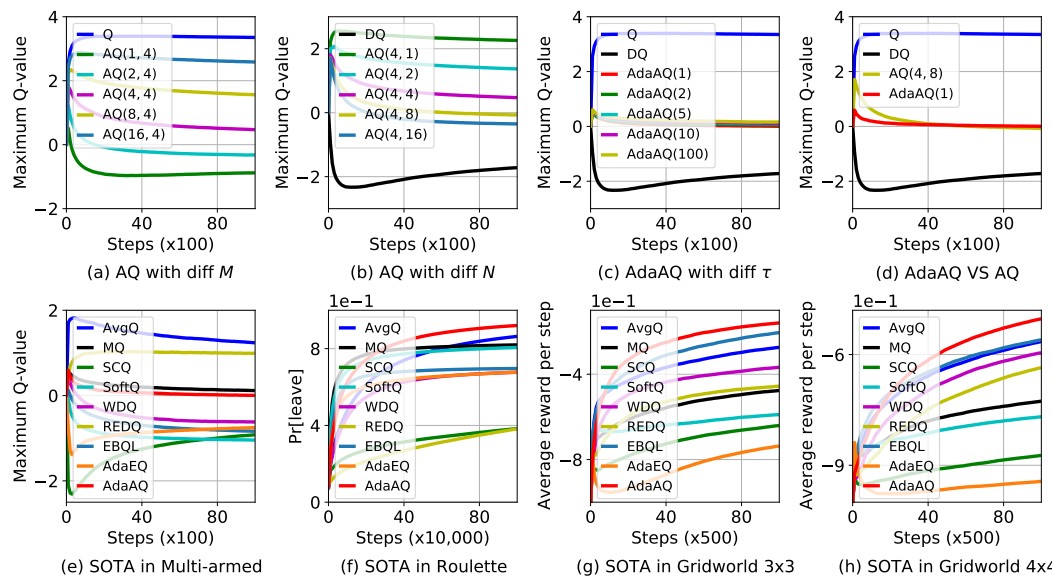

Figure 2: The performance of our AQ and AdaAQ in tabular MDP settings.

Table 1: The results of our AQ and AdaAQ with different parameters.

| Algorithm | Maximum Q-value | $\Pr[leave]$ | Average reward per step | |
|---|---|---|---|---|
| | Multi-armed bandit | Roulette | Gridworld $3 \times 3$ | Gridworld $4 \times 4$ |
| Q-learning | 3.35e0 | 40.83% | -8.55e-1 | -9.41e-1 |
| Double Q-learning | -1.72e0 | 26.15% | -6.65e-1 | -8.45e-1 |
| AQ(1,4) | -8.82e-1 | 49.88% | -5.35e-1 | -7.76e-1 |
| AQ(2,4) | -3.21e-1 | 83.73% | -5.16e-1 | -7.44e-1 |
| AQ(4,4) | 4.72e-1 | 87.43% | -4.92e-1 | -7.30e-1 |
| AQ(8,4) | 1.56e0 | 77.83% | -5.98e-1 | -7.74e-1 |
| AQ(16,4) | 2.59e0 | 56.04% | -7.33e-1 | -8.63e-1 |
| AQ(4,1) | 2.26e0 | 63.32% | -5.81e-1 | -7.97e-1 |
| AQ(4,2) | 1.37e0 | 82.70% | -5.27e-1 | -7.47e-1 |
| AQ(4,8) | -7.04e-2 | 85.69% | -5.80e-1 | -7.89e-1 |
| AQ(4,16) | -3.50e-1 | 78.26% | -5.98e-1 | -8.28e-1 |
| AdaAQ(1) | **4.14e-3** | 92.00% | -1.60e-1 | -5.04e-1 |
| AdaAQ(2) | 5.01e-3 | 90.53% | -1.57e-1 | -4.97e-1 |
| AdaAQ(5) | 9.32e-3 | 90.78% | -1.25e-1 | **-4.76e-1** |
| AdaAQ(10) | 1.08e-2 | 91.24% | **-1.20e-1** | -4.93e-1 |
| AdaAQ(100) | 1.61e-2 | **93.59%** | -1.50e-1 | -5.09e-1 |

## 5.2 DISCRETE-ACTION DRL EXPERIMENTS

We choose three discrete-action DRL experiment environments from PLE Urtans & Nikitenko (2018) and MinAtar Young & Tian (2019): Pixelcopter, Breakout, Asterix. Appendix G.1 provides the experiment settings. Figure 3 shows the average score per episode of our AdaADQN in discrete-action DRL settings, where the score is averaged over the last 100 episodes and the shaded area represents one standard error. One can observe that the average score curves of AdaADQN lie at the top, and are not sensitive to $\tau$. Appendix G.2 provides the table of comparison results between our AdaADQN and ten baselines, where all values are evaluated in the final round. More specifically, our AdaADQN improves the average score per episode over baselines by at least $\frac{37.89-30.13}{30.13} = 25.76\%$,

$\frac{13.89-10.64}{10.64} = 30.55\%$, and $\frac{16.13-9.77}{9.77} = 65.10\%$ in Pixelcopter, Breakout, and Asterix, respectively. In addition, Appendix G.3 provides the results of our ADNQ and AdaADQN with different parameters.

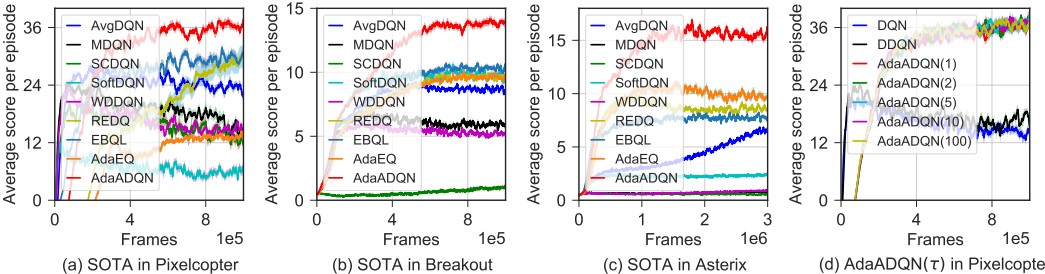

(a) SOTA in Pixelcopter    (b) SOTA in Breakout    (c) SOTA in Asterix    (d) AdaADQN($\tau$) in Pixelcopter

Figure 3: The average score per episode of our AdaADQN in discrete-action DRL settings.

### 5.3 CONTINUOUS-ACTION DRL EXPERIMENTS

We choose three continuous-action DRL experiment environments from Mujoco Brockman et al. (2016): Hopper, Ant, and Walker2d. Appendix H.1 provides the experiment settings. Figure 4 shows the average return of our AdaADDPG in continuous-action DRL settings, where the average return is averaged over the last 10 episodes. One can observe that the average return curves of AdaADDPG lie at the top, and are not sensitive to $\tau$. Appendix H.2 provides the table of comparison results between our AdaADDPG and five baselines, where all values are evaluated in the final round. More specifically, our AdaADDPG improves the average return over baselines by at least $\frac{3296.87-3033.64}{3033.64} = 8.68\%$, $\frac{3321.97-2818.44}{2818.44} = 17.87\%$, $\frac{4786.20-4248.04}{4248.04} = 12.67\%$ in Hopper, Ant, Walker2d, respectively. In addition, Appendix H.3 provides the results of our ADDPG and AdaADDPG with different parameters.

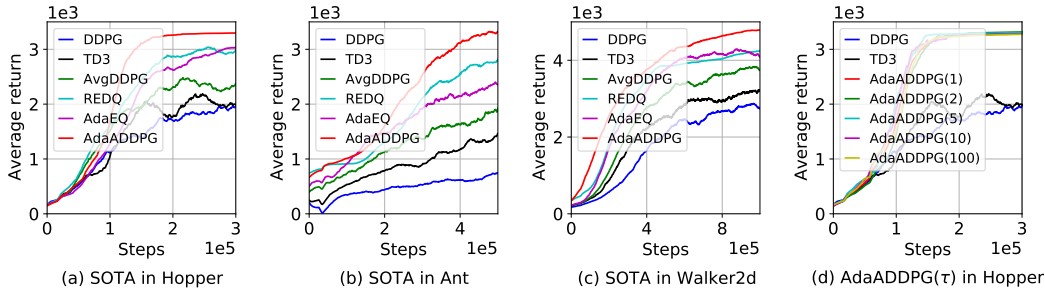

(a) SOTA in Hopper    (b) SOTA in Ant    (c) SOTA in Walker2d    (d) AdaADDPG($\tau$) in Hopper

Figure 4: The average return of our AdaADDPG in continuous-action DRL settings.

## 6 CONCLUSION

In this paper, we analyze the overestimation bias propagation process of Q-learning, and find that the prop-bias rather than the cur-bias dominates the maximum Q-value estimation bias. Then, we propose AQ, which breaks the unidirectional estimation bias propagation chains via alternately executing positive-negative bias algorithms. Based on AQ, we design an adaptive alternating strategy, leading to AdaAQ. More specifically, it applies a softmax strategy with the absolute value of TD error to determine whether AQ should execute Q-learning or Double Q-learning. We also extend AQ and AdaAQ to both discrete- and continuous-action DRL settings. Extensive experiment results show that our method outperforms several baselines drastically in tabular MDP, discrete-action DRL, and continuous-action DRL settings. More specifically, our method improves the average score or return over baselines by at least $65.10\%$ in Asterix and $17.87\%$ in Ant, respectively.

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
