# OpenReview forum: "Revisiting Overestimation Bias of Q-learning: Breaking Bias Propagation Chains Does Well"
_ICLR.cc/2026/Conference — ICLR 2026 Conference Withdrawn Submission_

### Official Review · Reviewer_H55i · 2025-10-20

**Soundness:** 2
**Presentation:** 2
**Contribution:** 2
**Rating:** 4
**Confidence:** 3

**Summary:**

This manuscript analyzes the overestimation bias problem in the Q-learning algorithm and points out that the bias propagation chain is the main factor slowing convergence. The authors theoretically prove that two appropriate alternating parameters can eliminate the propagation bias, and based on this, they design an adaptive alternating strategy, which is validated through experiments on discrete-action and continuous-action DRL tasks.

**Strengths:**

1. The overall structure of the manuscript is clear.

2. It provides theoretical insights into the error propagation chain.

3. The writing quality is very good.

**Weaknesses:**

1. The propagation is based on the current bias (Theorem 1). When the current bias is large or small, the cumulative error also becomes high or low. Since single and double Q-learning essentially reduce the current overestimation or underestimation, the proposed method is, in essence, still addressing the current estimation bias. From my perspective, the error propagation chain is therefore a result rather than a cause.

2. The approach of alternately using single Q-learning (positive bias) and double Q-learning (negative bias) is not elegant.

3. The research focus seems somewhat outdated and of limited value. Moreover, the proposed method does not appear to offer significant insights or inspiration for future developments in the community. The use of older baseline algorithms also reflects this issue.

**Questions:**

The authors only need to provide rebuttals to the weaknesses.

---

### Official Review · Reviewer_UMkK · 2025-10-31

**Soundness:** 3
**Presentation:** 3
**Contribution:** 2
**Rating:** 4
**Confidence:** 3

**Summary:**

The paper looks at why basic Q-learning tends to overestimate how good actions are and argues that the bigger issue isn’t just the current step’s optimism, but how that bias keeps getting passed forward and compounded over time. Building on this view, the authors propose Alternating Q-learning (AQ), which periodically switches between a method that’s optimistically biased (standard Q-learning) and one that’s pessimistically biased (Double Q-learning) to cancel out the bias as it propagates. They then make the switching adaptive (AdaAQ): for each state–action pair, the algorithm uses a softmax over absolute TD errors to decide whether to update with Q-learning or Double Q-learning. They analyze the bias composition theoretically and claim there exist alternating schedules that eliminate the propagation bias. Finally, they extend the idea to deep RL (DQN/DDQN; DDPG/TD3) and report improvements on MinAtar/PLE games, and MuJoCo tasks.

**Strengths:**

Importance. The paper tackles a central issue in RL.

Novelty. The proposed method seems to be new.

Technical aspect. It looks interesting to formalize the bias as “current vs. propagation,” prove a decomposition (Theorem 1) and provide conditions where propagation dominates (Corollary 1); it seems mathematically tidy and easy to follow.

Clarity. The presentation is reasonably clear.

**Weaknesses:**

Some important claims are not well-supported. See below.

Your main conceptual claim is that propagation bias, not current bias, dominates and slows convergence; beyond Corollary 1 and the simple bandit example (Fig. 1), can you quantify this dominance across a wider set of MDPs (e.g., different reward scales, γ, and action counts) and show how the ratio evolves in more realistic environments?

The decomposition (Theorem 1) leans on Assumption 1 (uniform i.i.d. Q-errors with shrinking support); how sensitive are your conclusions and the AQ/AdaAQ benefits to violations of this (e.g., heavy-tailed noise, correlated function-approximation errors, non-stationarity from replay and target networks)? Can you relax or empirically stress-test this assumption?

AdaAQ switches based on a softmax over |TD-errors| from Q vs. Double Q updates (Eq. 6); why should absolute TD error be the right proxy for the sign/magnitude of estimation bias rather than, say, a calibrated uncertainty estimate, an advantage-gap signal, or an online bias estimator? Can you ablate this choice and compare against alternative switching signals?

Note that every design choice should be well-supported.

Since bias and action gaps vary across states, would a per-state or per-feature schedule for M,N (or a learned gating network) outperform the global softmax rule?

In the tabular and deep experiments, some ensemble-style baselines (e.g., REDQ, EBQL) depend heavily on ensemble size and target update cadence; can you provide a thorough hyperparameter sweep summary and compute-matching (updates per environment step) to ensure fairness, especially where AdaAQ shows large margins?

Since the pitch is about propagation bias, do you have a direct empirical estimator/diagnostic of propagation vs. current bias through training (beyond the bandit toy in Fig. 1), perhaps via controlled MDPs where ground-truth Q is known, or via synthetic noise injections to trace how early bias moves through bootstrapping?

In deep RL, max-operator bias interacts with target networks, replay, and non-linear approximation error; can you isolate which part AdaADQN/AdaADDPG helps most (e.g., ablate target-network lag, noisy layers, or exploration) and whether alternating mainly reduces over- or under-estimation in practice?

**Questions:**

see above.

---

### Official Review · Reviewer_TVN7 · 2025-10-31

**Soundness:** 3
**Presentation:** 3
**Contribution:** 3
**Rating:** 6
**Confidence:** 4

**Summary:**

This paper proposes alternating Q-learning, for mitigating over-estimation in Q-learning with function approximation. Many approaches that mitigate over-estimation, particularly Double Q-learning, end up causing under-estimation. This work specifically balances between this under-estimation and over-estimation by switching between Q-learning and double Q-learning during learning. The expected estimation bias of this algorithm is derived. The paper further proposes a heuristic to automate the frequency of this switching to avoid having to tune the frequency hyperparameters. The proposed method is easily applied to many value based RL algorithm; they propose applications to several state of the art deep RL algorithms. When using this method, the deep RL algorithms outperform the standard variants in a variety of RL tasks, including tabular, image based, and continuous control RL settings.

Overall this paper presents a promising and simple method, but some clarifications on the theory and experimental results are needed (outlined below). Specifically, one issue is that there is a claim on curr-bias that may not be strictly true. Additionally, there are some issues with showing variability across runs (either not having standard errors or CIs, or having strangely narrow standard errors). I have already set the ratings assuming these can largely be addressed, but with further clarification on these points, I would be happy to increase my ratings further.

**Strengths:**

The paper presents an extensive variety of experiments, using a variety of baseline environments and benchmarks along with 20 random seeds per algorithm allowing for reasonable conclusions to be drawn about the comparisons between algorithms.

A detailed and thorough study of the literature on over-estimation in Q-learning is presented in suitable detail to motivate the necessity of further methods to address the shortcomings of current methods.

The paper presents a reasonable and easy to follow justification for their choice of heuristic in the adaptive alternating Q-learning, and demonstrates clearly that this heuristic is better than any fixed choice of hyperparameters across environments.

**Weaknesses:**

On line 273/274, the paper states that when Double Q-learning is used to update, the curr-bias is negative. This fact relies on the conclusion that Double Q-learning under-estimates, which can be shown for an unbiased function approximator, as outlined as an assumption in Lemma 1 of the original work (Hasselt, et al 2010). In this work it is assumed that the function has a bias \mu_t, which may not be 0. This might mean that lemma 1 from Hasselt et al, (2020) cannot be invoked. A clarification on why the curr-bias is still negative under assumption 1 is needed to support the claim that there exists an (M,N) which eliminates the prop-bias. The abstract claims “We show theoretically that there exist two suitable alternating parameters to eliminate the propagation bias”, which may not be strictly true.

The experiments are quite thorough and the authors clearly put quite a bit of care into the design and execution. There are two concerns, however, that should be addressed.
First, in the continuous action DRL experiments, there does not seem to be standard errors or confidence intervals in the graphs. The appendix says that 20 random runs were used, which means they could be computed. It is hard to conclude that ADaADDPG outperforms the other algorithms without this measure.

Second, for the arcade learning environment (ALE) experiments, the standard errors seem very small in comparison to other works using the ALE. One possible reason for this is that the standard error might be assuming 100 samples: 10 for the 10 rollouts times 10 for the number of runs. Instead, what should be done is that the 10 rollouts give an estimate of the return for the policy in one run, and the CI or stderror uses 10 samples for the 10 runs. This is just a guess, of course it could be for other reasons. Can you explain why these standard errors are so small?

Finally, I think there could be more discussion on the comparison to max-min Q learning (cited by the authors). This paper under review explains that the max-min Q-learning work also has an approach to go from underestimation to overestimation, and with an appropriate choice, can balance between the two. Does this approach already solve this issue? If not, why not?

(Putting Minor Points here, so there is no separate box. These are not major issues)
1. Each subsection should start with a small preamble to explain what the subsection is about.
2. For the theory portion of the paper, the introduction of assumption 1 could be clarified, since it appears currently to be grouped among the common assumptions made by other cited papers. The assumption that, due to previous Q-learning steps, the function approximator itself now over-estimates Q values is not the same assumptions made by Thrun and Schwartz (1993), or Lan et al (2020) and therefore should be separated from the other common assumptions. This assumption is reasonable in my view, but is a separate assumption from the one made in these cited papers.

**Questions:**

1. Why is the curr-bias from double Q-learning still negative under assumption 1?

2. How were the standard errors for the ALE experiments computed, since they appear  smaller than expected?

3. Why do previous method that balance between under-estimation and over-estimation (e.g., max-min Q-learning) not resolve the issue being tackled by this paper? (I am not doubting that we need the new algorithm in this paper, but rather am asking for a more explicit placement relative to existing solutions).

---

### Note · Authors · 2025-11-13

I have read and agree with the venue's withdrawal policy on behalf of myself and my co-authors.